# Ion Mobility Spectrometry in Food Analysis: Principles, Current Applications and Future Trends

**DOI:** 10.3390/molecules24152706

**Published:** 2019-07-25

**Authors:** Maykel Hernández-Mesa, David Ropartz, Ana M. García-Campaña, Hélène Rogniaux, Gaud Dervilly-Pinel, Bruno Le Bizec

**Affiliations:** 1Department of Analytical Chemistry, Faculty of Sciences, University of Granada, Campus Fuentenueva s/n, E-18071 Granada, Spain; 2INRA, UR1268 Biopolymers Interactions Assemblies, F-44316 Nantes, France; 3Laboratoire d’Etude des Résidus et Contaminants dans les Aliments (LABERCA), Oniris, INRA UMR 1329, Route de Gachet-CS 50707, F-44307 Nantes CEDEX 3, France

**Keywords:** food quality, IMS, food composition, food process control, food authentication, food adulteration, food safety

## Abstract

In the last decade, ion mobility spectrometry (IMS) has reemerged as an analytical separation technique, especially due to the commercialization of ion mobility mass spectrometers. Its applicability has been extended beyond classical applications such as the determination of chemical warfare agents and nowadays it is widely used for the characterization of biomolecules (e.g., proteins, glycans, lipids, etc.) and, more recently, of small molecules (e.g., metabolites, xenobiotics, etc.). Following this trend, the interest in this technique is growing among researchers from different fields including food science. Several advantages are attributed to IMS when integrated in traditional liquid chromatography (LC) and gas chromatography (GC) mass spectrometry (MS) workflows: (1) it improves method selectivity by providing an additional separation dimension that allows the separation of isobaric and isomeric compounds; (2) it increases method sensitivity by isolating the compounds of interest from background noise; (3) and it provides complementary information to mass spectra and retention time, the so-called collision cross section (CCS), so compounds can be identified with more confidence, either in targeted or non-targeted approaches. In this context, the number of applications focused on food analysis has increased exponentially in the last few years. This review provides an overview of the current status of IMS technology and its applicability in different areas of food analysis (i.e., food composition, process control, authentication, adulteration and safety).

## 1. Introduction

In the current context of food trade globalization and due to the recognized impact of the diet on human health, food analysis has become more important than ever. Food analysis has today gone beyond the traditional analysis of the major components of food and is more complex and broader. In addition to the nutritional value of foodstuffs (i.e., carbohydrates, proteins, lipids, vitamins, minerals and water), food analysis has been focused on food safety for a long time, mainly in the determination of residues of pesticides and veterinary drugs. Nowadays, food safety analysis encompasses a wide variety of compounds including natural contaminants (e.g., toxins) or anthropogenic contaminants such as persistent organic pollutants (POPs) [1]. Understanding the interactions of food with the environment and consequences (i.e., large-scale production, organic production, environmental contamination, etc. and including the control of food processes, packaging, etc.), as well as its effects on consumers (e.g., investigation of bioactive compounds), is also gaining great importance. Other issues such as food authentication (i.e., quality, origin, etc.), adulteration and fraud detection have also acquired great relevance in the field of food chemistry in the last few years [2].

Food analysis involves the determination of a wide range of compounds with different chemical nature. Chromatographic techniques (i.e., liquid chromatography (LC) and gas chromatography (GC)) coupled to mass spectrometry (MS) are the gold standard for this purpose because they allow the analysis of molecules with different polarity and volatility as well as provide mass spectra for compound identification. However, LC–MS and GC–MS methods still face several challenges related to the complexity of food matrices, the presence of compounds at different concentration levels (from pg/kg and pg/L to mg/kg and mg/L levels), and the existence of isobars and isomers that are not separated in the chromatographic dimension and cannot be distinguished by MS.

Consequently, the development of more advanced analytical strategies is required for the analysis of food composition, including nutritive and bioactive components, as well as to guarantee food safety and avoid food fraud. Within this framework, ion mobility spectrometry (IMS) has been recently introduced in the food chemistry field in order to improve LC–MS and GC–MS workflows. Despite the fact that IMS can be used alone as analytical tool, it can also be coupled to other analytical separation techniques such as LC, GC, capillary electrophoresis (CE), or supercritical fluid chromatography (SFC) as well as to MS, enhancing their performance characteristics in terms of sensitivity, peak capacity, and compound identification [3,4]. Its coupling with front-end separation techniques and with MS has emerged as a useful approach to extend the current boundaries of analytical methods in food science, and it can be anticipated that it will rapidly be growing in this field.

The implementation of IMS within the food analytical field is quite new [5,6,7,8], and is still barely known by many researchers in this scientific area. Despite IMS fundamentals have been developed since the beginning of the 20th century [9], it has not been until the recent commercialization of hyphenated ion mobility-mass spectrometry (IM–MS) instruments when this technique has really caught the attention of researchers from multiple fields, including food science. As a result, the number of publications on IMS applications in food analysis has rapidly increased over the last few years (Figure 1). Within this context, this review provides a general overview of IMS principles and presents the current state of the art of this technology for food analysis purposes.

## 2. Overview of Ion Mobility Spectrometry (IMS) Technique and Potential

IMS is an electrophoretic separation technique in which ionized compounds are separated in a neutral gas phase at atmospheric or near to atmospheric pressure. Therefore, separation takes places under an electric field (E) and is the result of the difference in mobility (K) of ions in the drift cell. K refers to the time (t_d_) required by the ions to traverse the length (l) of the mobility cell and is related to the electric field according to Equation (1). In this equation, v_d_ represents the steady-state net ion/gas relative velocity [10].

(1)$$K=\;\frac{l}{t_{d\;}E}=\frac{v_{d}}{E}$$

This physical property depends on several experimental conditions such as temperature (T), pressure (p) and gas number density (N). The reduced mobility (K_0_), which refers to standard conditions (Equation (2); N_0_ = 2.687 × 10^25^ m^−3^, p_0_ = 760 Torr, T_0_ = 273.16 K), is typically reported instead of K in order to allow comparison between studies independently of the experimental conditions.
(2)$$K_{0}=K\;\frac{N}{N_{0}}=K\frac{p}{p_{0}}\frac{T_{0}}{T}$$

Although the ‘momentum transfer collision integral’ (Ω), commonly referred to as the collision cross section (CCS), is reported as the response resulting from ion mobility measurements [11], t_d_ or K are actually the variables that are measured when performing IMS experiments. Nevertheless, both parameters can be easily correlated according to Mason–Schamp equation (Equation (3)) when the separation occurs at low electric fields [12].
(3)$$K=\;\frac{3}{16}\frac{\mathrm{ze}}{N}{\left(\frac{2\pi }{{µk}_{B}T}\right)}^{1/2}\frac{1}{Ω}$$
where z and e represent the absolute charge of the ion and the elementary charge, respectively; µ encompasses the reduced mass of the ion–neutral drift gas pair (i.e., μ = mM/(m + M); m and M are the ion and gas-particle masses, respectively); and k_B_ is the Boltzmann constant.

CCS represents the averaged momentum transfer impact area of the ion and is a molecular parameter related to ions size, shape and charge state. Therefore, CCS is widely used for structural elucidation since it provides knowledge about the three dimensional (3D) conformation of ions in the gas phase [13,14]. The correlation existing between this parameter and the mass-to-charge ratio (*m*/*z*) is not negligible. However, it also gives additional information to retention index (e.g., retention time, electrophoretic mobility, migration time, etc.), mass spectra, fragmentation and isotopic patterns, etc., for peak annotation in analytical workflows, especially in omics approaches (e.g., metabolomics, lipidomics) and, ultimately, for compound identification [15,16].

### 2.1. IMS Instrumentation

Nowadays, there is a wide variety of stand-alone IMS and IM–MS instruments on the market, but they are based on different technologies that offer different advantages [4,14,17]. They can be classified in time-dispersive, space-dispersive and trapping (i.e., ion confinement and release) technologies [18]. Drift tube ion mobility spectrometry (DTIMS) and travelling tube ion mobility spectrometry (TWIMS) are two different types of time-dispersive forms. For example, DTIMS is currently commercialized by Agilent (i.e., 6560 Ion Mobility LC/quadrupole-time of flight (Q–ToF)), Excellims Corporation (i.e., high-performance ion mobility spectrometry (HPIMS) systems including MA3100 and RA4100 HPIMS–MS instruments), Gesellschaft für Analytische Sensorsysteme mbH (G.A.S.) (i.e., GC–IMS systems) and TOFWerk (i.e., IMS-ToF), whereas TWIMS is available from Waters Corporation (i.e., Synapt G2-Si and Vion IMS QTof). High-field asymmetric waveform ion mobility spectrometry (FAIMS) and differential ion mobility spectrometry (DIMS or DMS) belong to space-dispersive techniques. Both FAIMS and DMS are based on the same principles of operation and mainly differ in the geometry of the cell. FAIMS cells are curved whereas DMS cells present a planar geometry and, consequently, they can lead to different analytical properties [19]. FAIMS systems are currently available from Owlstone Medical (i.e., ultraFAIMS) and Thermo Fisher Scientific (i.e., Thermo Scientific FAIMS Pro interface), whereas DMS is commercialized by SCIEX (i.e., SelexION). Until now, trapped ion mobility spectrometry (TIMS), which represents a type of trapping technology, is the only IMS system of its class that is currently commercially available (i.e., timsTOF from Bruker Daltonics).

Regarding CCS-related measurements, they can only be carried out using IMS instruments that operate at low electric fields (e.g., DTIMS) because the reduced mobility is independent of the electric field under this condition. K_0_ becomes dependent of the reduced field strength (E/N) at high electric fields (E/N > 4–10 Townsends (Td); 1 Td = 10^−21^ V m^2^) and Equation (3) is no longer applicable [10]. In this sense, only primary DTIMS methods can be applied to obtain CCS values directly, whereas secondary DTIMS, TWIMS and TIMS methods require system calibration using reference compounds of known CCS. Only those compounds characterized in terms of CCS by primary DTIMS methods can be used as calibrants [20]. The same results should be obtained for the CCS of a specific ion independently of the IMS platform employed if secondary methods operate under the same experimental conditions (i.e., temperature, drift gas, E/N, etc.) as primary methods. Since this is not always possible, CCS is a method-dependent value [20]. Specific annotation (^IMS form^CCS_drift gas_, IMS form: DT for DTIMS, TW for TWIMS, TIMS for TIMS; and drift gas: N_2_, CO_2_, He, etc.) is currently accepted to indicate the type of IMS technology and drift gas used for CCS measurements [11].

In DTIMS and TWIMS methods, ions travel through the drift cell against the buffer gas describing a similar path, so they are separated according to their mobility in the drift cell (Figure 2). Compact molecules collide less frequently with the molecules of drift gas (normally N_2_ or He, but also CO_2_), so they present a higher mobility (i.e., a smaller CCS) and cross the cell faster than elongated molecules.

DTIMS is the former and simplest form of IMS, and DTIMS cells consist of a series of piled electrodes that generate a weak uniform electric field. In general, E/N varies between 1 to 15 Td [20]. DTIMS works as a primary method when stepped field experiments are carried out. In this case, the arrival time (t_A_) is measured at multiple fixed drift voltages, and each t_A_ value represents the sum of drift time (t_d_) of ions and the time elapsing between the moment when ions exit the drift tube and their detection (t_0_). Subsequently, the reduced mobility can be directly obtained by applying Equation (4) and the CCS is further calculated by Mason–Schamp equation, Equation (3) [21].

(4)$$t_{A}=t_{d}+t_{0}=\left(\frac{l^{2}}{K_{0}}\times \frac{T_{0}p}{{\mathrm{Tp}}_{0}}\right)\;\times \frac{1}{\Delta V}=\mathrm{slope}\times \frac{1}{\Delta V}+t_{0}$$

In general, the term ‘arrival time distribution’ (ATD) should be used because IMS measurements are based on a population of ions rather than on a single ion and they do not reach one point of the instrument at the same time. ATD gives information about how homogeneous is the population of ions, and it should be reported in addition to t_A_ [20]. Furthermore, it must be noted that stepped field methods are limited to the K_0_/CCS characterization of molecules because the analysis of complex samples is not feasible with them. Consequently, DTIMS usually works in single-field mode in which a single linear voltage is applied along the drift tube. In this condition, DTIMS operates as a secondary method and CCS calibration is required to measure this molecular descriptor [22].

Unlike in DTIMS, a dynamic electric field is applied in TWIMS to separate the ions [17]. TWIMS systems consist of a stacked ring ion guide where a pulsed differential voltage is applied to each electrode. As a result, a wave of electric potential travels along the drift tube and propels the ions axially. Ions are subjected to a varying voltage with a maximum E/N of 160 Td [10], and their separation depends on the speed and magnitude of the voltage wave. Voltage waves move faster than the ions, so they roll over the wave and require a succession of them to reach the exit of the mobility cell. In addition, in order to prevent the ions being pushed towards the drift tube wall, radio-frequency (RF) voltages of opposite phases are periodically applied to adjacent electrodes causing the radial confinement of ions [4,23].

On the other hand, TIMS instruments consist of three regions of electrodes: entrance funnel, TIMS tunnel, and exit funnel (Figure 2). Unlike in DTIMS and TWIMS where an electric field is applied to push the ions through the cell, in TIMS systems, an electric field (i.e., 45–150 Td) is applied to trap the ions and they are only dragged through the drift tube towards the detector by the gas [24]. Initially, ions are accumulated for a fixed period of time and directed to the entrance funnel. Subsequently, they are released into the TIMS funnel traversing an axial electric field gradient (EFG). When the velocity of the buffer gas (v_g_) is equaled by the opposite steady state drift velocity of the ion (v_d_), ions reach a stationary state and are trapped in a moving column of gas. Finally, the EFG is decreased and ions are released towards the exit funnel. Large ions with lower K are eluted before more compact ions since the dependence of the drift time on K_0_ is opposite to DTIMS [25].

Finally, FAIMS and DMS instruments do not separate ions in function of their mobility in a neutral gas as in DTIMS, TWIMS or TIMS, but rather by the ratio of low-field to high-field mobilities [17]. Consequently, CCS values cannot be obtained by these systems. In FAIMS/DMS systems, a time-dependent electric field is applied between two parallel electrodes. Low and high electric fields of opposite polarities are simultaneously applied (i.e., high electric fields > 30 Td) [26], taking into account that the product of the voltage (commonly referred as dispersive voltage, DV) and time for each condition must be the same as shown in Figure 2 [19]. Low electric fields are applied for longer periods than high electric fields. Moreover, a compensation voltage (CV) is applied to one of the electrodes with the aim of avoiding that ions collide against them. As a result, the trajectory of ions is altered ensuring that they migrate to the exit of the cell. In this sense, FAIMS and DMS act as selective instruments because the CV is an analyte-dependent parameter and they are not able to scan the complete CV range during the transition of ions through the cell. Unlike in other IMS forms where all the ions are normally detected, in FAIMS/DMS technology, only those analytes related to the selected CVs reach the detector.

More detailed information about the operation and physical principles of the different IMS forms can be found in specialized literature [17,19,25,27,28]. Furthermore, a guide about how IMS experiments must be reported, including CCS measurements, has been recently published [20]. It is recommended to follow this guide for the communication of IMS-related results because it will probably set the basis for future guidelines and standards in ion mobility.

### 2.2. Collision Cross Section (CCS) for Structural Elucidation

As previously mentioned, IMS also gives access to the CCS characteristic which, in certain cases, provides additional information to mass spectra and retention index. It can be highly valuable for a higher confidence in the determination of residues and contaminants in food safety or for achieving a more complete fingerprint of food products. In the last few years, several CCS databases have been built in an attempt to use this characteristic as an identification parameter [11,15]. In general, there is still much controversy about the added value provided by this parameter in comparison to mass spectra. It cannot be denied that CCS and *m*/*z* are not fully orthogonal parameters since a close correlation exists between them, as observed in Figure 3B. Nevertheless, slight differences existing between the CCS of molecules with similar or equal *m*/*z* (i.e., isobars and isomers) can be enough to distinguish them (differences of at least 1.5–1.8% (2.0–6.5 Å^2^) in apparent CCS values for accurate CCS determination [29]).

Regarding the application of CCS databases, an error threshold of ±2% is currently accepted for CCS measurements in comparison to CCS values reported by them (Equation (5)). However, the threshold of ±2% could potentially be reduced, which will give more confidence to the results [30]. Even so, this threshold for CCS measurements can still result in being more effective than isotopic pattern or fragmentation criteria to reduce the number of false positive results in automated screening workflows and avoid the requirement of post manual verification or confirmation analysis [31].
(5)$$\%\;\mathrm{CCS}\;\mathrm{error}=\;\frac{\left({\mathrm{CCS}}^{\mathrm{measured}}\;-{\mathrm{CCS}}^{\mathrm{database}}\right)}{{\mathrm{CCS}}^{\mathrm{database}}}\;\times 100$$

### 2.3. IMS Hyphenation

IMS separations typically take place in the millisecond range, so they can be easily carried out after traditional chromatographic or electrophoretic separations (i.e., LC, GC, SFC, and CE), which occur in the second range. The selection of the chromatographic/electrophoretic technique is obviously influenced by the nature of analytes (e.g., SFC for polar compounds, CE for polar and ionic compounds, GC for volatiles). Moreover, MS separations last microseconds so IM-MS hyphenation is also possible. IMS is normally coupled to time-of-flight (ToF)-MS due to its high acquisition rate. Indeed, as indicated in Section 2.1, several IMS systems combined with ToF-MS technology are already available on the market as integrated instruments.

GC was one of the first analytical techniques to be coupled to IMS, which has been mainly used as a detector in GC–IMS configurations [3,32]. GC–IMS hyphenation has been widely applied to the analysis of volatile compounds in food samples because stand-alone IMS systems generally provide low resolution. Both analytical techniques present a high degree of orthogonality, so selectivity and peak capacity are improved. However, LC–IM–MS platforms are currently becoming very popular and are used for numerous applications [4]. In this context, the integration of IMS in LC–MS workflows introduces a third separation dimension that improves peak capacity and allows the separation of isobars and isomers [33]. The implementation of IMS in LC–MS workflows increase peak capacity at least 2 or 3-fold [34,35], although this improvement ultimately depends on IMS resolution and the analytical application. Moreover, chromatographic peaks are extracted from background noise which provides cleaned-up chromatograms and mass spectra, leading to an improvement of the limits of detection (LODs) and sensitivity. Mass spectral data can also be interpreted more easily. Undoubtedly, these characteristics have contributed to the implementation of IMS technology in the food science field where samples present a high complexity and some compounds need to be determined at low concentration levels (i.e., ppb and ppt ranges).

From a technical point of view, limited dynamic range and ion loss have been traditionally attributed to LC–IM–MS in comparison to LC–MS methods [36]. However, several technological advances have been accomplished in the last years and, for example, some recent studies indicate that hyphenation with an ion mobility device does not affect the linear dynamic range [37]. Furthermore, it is important to remark that each IMS form can improve the performance of LC–MS methods in one way or another, but they are not exempt of limitations [4,14]. Therefore, the choice of the IMS technology will depend on the purpose of the intended application. In general, FAIMS and DMS provide higher orthogonality to MS than DTIMS and TWIMS but they act as signal filters according to the CV selected. Consequently, information about the sample is lost because the number of detected compounds is limited.

Finally, independent of whether stand-alone IMS, IM–MS or LC–IM–MS platforms are used, samples need to be ionized prior to entering into the mobility cell. Nowadays, electrospray ionization (ESI) is the gold standard for IMS analyses, although ^63^Ni radioactive ionization sources are widely used in portable IMS instruments. Other ionization sources such as photoionization, corona discharge (CD) and pulse glow discharge (PGD) have also been used in food applications, but to a lesser extent [8]. Sample ionization is a process to be controlled because, as well as ion transportation and storage, it influences the conformation of ions and, consequently, their mobility. In addition, different adducts and ions with different charges states and sites (e.g., protomers) can be detected for the same compound [20], and their formation during the ionization can be altered by experimental conditions (i.e., ionization source and related parameters, solvents composition, etc.). In the context of food analysis, specifically in food safety, the identification of protomers is of high relevance since it could justify why some compounds lead to non-compliance with confirmation criteria based on ion fragmentation ratios and are not detected in screening analyses [38,39]. In addition, if it is applicable, a different t_d_, K or CCS could potentially be obtained for each ion which provides more information for peak annotation. But it also hinders data treatment due to the number of species or ‘features’ that are detected. In this sense, data treatment generally remains as the main bottleneck of hyphenated LC–IM–MS systems, and a higher development is still required to successfully integrate IMS data in current LC–MS workflows.

## 3. Applications of IMS in Food Analysis

The application of IMS in food science is still in its early years, especially those approaches involving IM–MS or LC–IM–MS hyphenation. Despite IMS, and specifically IM-MS, has found its main application in omics approaches (i.e., proteomics and metabolomics, including lipidomics and glycomics) [40,41,42], this technique has barely been exploited for the specific analysis of composition and nutritional value of food, and their health benefits and risks (i.e., foodomics). Nevertheless, it is expected that its use will be extended rapidly to foodomics where a wide range of food metabolites need to be characterized for understanding their effects on human health. For example, the effects of coffee consumption on lipids profile (i.e., 853 lipid species from 14 lipid classes) have been investigated by DMS-triple quadrupole (QqQ)/linear ion trap (LIT)–MS [43]. This lipidomics approach has suggested that coffee intake alters glycerophospholipid metabolism and supports previous studies about the health benefits of drinking coffee.

The interest for this technology is growing and an increasing number of IMS-based methods have already been developed for food analysis, especially for food safety and food authentication purposes. A selection of the most recent applications of IMS in food science, including food composition, process, authentication, adulteration, and safety, is presented below. This section intends to provide a general vision of the potential of this technique for food analysis rather than being a comprehensive summary of all related articles that have been reported in the last years.

In general, IMS applications in food science cover from the determination of a predetermined number of compounds (i.e., targeted analysis) to the analysis of a large non-predetermined set of molecules (i.e., non-targeted analysis). As will be shown in the following sections, IMS has mainly been applied to targeted or semi-targeted analysis until now. However, it is expected that it will be widely used in non-targeted approaches, such as in food fingerprinting, where a large number of compounds are detected and the performance characteristics of analytical methods need to be improved (i.e., requirement of higher resolving power, tools to support molecular identification, simplification of mass spectra, etc.). On the other hand, current IMS applications in food analysis can be classified in three types: 1) those using stand-alone IMS instruments in which their intrinsic low selectivity is not a limitation, 2) approaches where IMS is coupled to a chromatography technique, usually GC, and mainly acts as a detector, and 3) applications based on LC–IM–MS workflows (alternatively GC–IM–MS) in which the potential of IMS is fully exploited [i.e., extra separation dimension that improves selectivity and sensitivity (separation of isomers and isobars, and isolation of compounds of interest form background noise), compound selection based on their CVs (in FAIMS and DMS), CCS as an additional molecular descriptor (except in FAIMS and DMS)].

### 3.1. Food Composition

From a chemical composition point of view, food consists of a wide variety of compounds that can be divided in macronutrients (i.e., proteins, carbohydrates, lipids and vitamins) and micronutrients (e.g., polyphenols, etc.). Nowadays, IMS applications in food composition cover a wide variety of substances (e.g., lipids, peptides, phenolic compounds, terpenes, etc.) [44,45,46,47], including allergens [48], and samples (e.g., flours, olive oil, mushrooms, etc.) [48,49,50]. Due to the complexity of food matrices, stand-alone IMS has rarely been investigated in food composition studies, although it has found a wider application in food process analysis where few and specific compounds are determined, as will be shown in Section 3.2. As an example, stand-alone IMS has been used for the analysis of seven alcohol sweeteners in chewing gum, identifying the presence of sorbitol [51]. Other peaks were also observed in the ion mobility spectrum which were attributed to gum base components. However, this fact highlights the requirement of IM–MS for proper identification. Furthermore, more analytical information is obtained when IMS is combined with other analytical techniques such as LC or GC. Chromatographic and IMS separations are not correlated [44,46] (Figure 3), so their coupling improves peak capacity and enables the detection of a larger number of compounds. This advantage is widely used for food fingerprinting at low cost [50,52,53], without applying MS which is an expensive technology. In this case, if complete food characterization is required, standards of those substances that are potentially present in the sample must also be characterized in terms of retention time and mobility, drift time, or CCS. Consequently, peaks resulting from sample analysis can be tentatively identified (MS is mandatory for identity confirmation).

**Figure 3 molecules-24-02706-f003:**
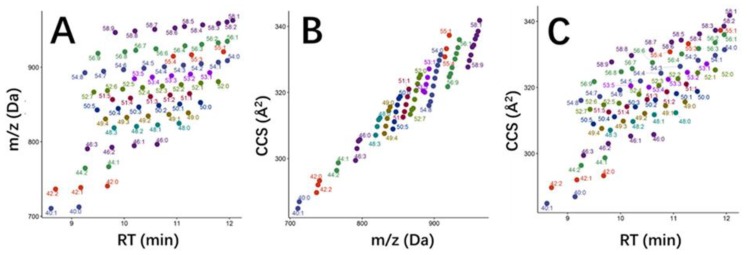
Selected triacylglycerides found in milk and comparison of the measured orthogonal parameters: (**A**) *m*/*z* vs. retention time (RT), (**B**) collision cross section (CCS) vs. *m*/*z*, and (**C**) CCS vs. RT. Figure reprinted with permission from [44]. Copyright (2018) American Chemical Society.

In this context, the volatile organic compounds (VOCs) fraction of olive oils has been widely studied by GC–IMS, and headspace (HS) analysis has been usually carried out for this purpose [49,52,53]. In general, IMS offers high sensitivity allowing the detection of olive oil volatiles in the range of 0.01–0.05 ppm, which is usually below the odor threshold of many compounds in extra virgin olive oils (EVOO) [53]. Several aldehydes (e.g., hexanal, 2-hexenal), ketones (e.g., 1-penten-3-ol), alcohols (e.g., 1-pentanol, 1-hexanol, etc.) and esters (e.g., hexylacetate), which are related to desirable attributes of olive oils, are present in their VOCs fraction. Their targeted analysis by HS–GC–DTIMS combined with chemometrics has been shown to be an effective strategy to characterize olive oils and distinguish them according to their quality (i.e., extra virgin, virgin and lampante) [49]. As alternative to HS analysis, olive oils samples can be submitted to laser desorption–GC–IMS analysis for the simultaneous characterization of VOCs fraction and detection of semi- and non-volatile compounds (e.g., (*E*,*Z*)-2,4-dodecadiene, 1-dodecene, (*E*)-2-hexenal, phenol, ß-pinene, benzaldehyde, acetic acid, limonene, 1-hexanol, nonanal, 1-heptanol, and octanal [54]. In comparison to HS, a higher number of signals were detected when laser desorption was applied to the analysis of oil samples. Despite the analysis of certain markers providing useful information about olive oils quality, statistical models for olive oils classification are improved when oil fingerprinting (i.e., semi-targeted or non-targeted analysis) is carried out [52,53]. This is because food fingerprinting gives a more complete picture of samples composition than the analysis of specific compounds. In the case of using GC–IMS methods, this implies that samples are differentiated according to biomarkers not identified, but characterized analytically. In addition to olive oils, HS–GC–DTIMS has also been applied to characterize the volatile fingerprint of fresh and dried *Tricholoma matsutake* Singer samples [50] and green tea aromas [55].

Unlike GC–IMS, LC–IMS coupling has not been evaluated so far for food composition analysis according to the literature. Nevertheless, the determination of phenolic acids in seedling roots by high-performance liquid chromatography (HPLC)–ESI–DTIMS has been recently reported [56], and this application can be extended to food composition analysis. Several compounds co-eluted in the chromatographic dimension (i.e., vanillic acid and caffeic acid; ferulic acid and sinapic acid; benzoic acid and salicylic acid), but they were separated in the ion mobility dimension. It demonstrated the potential of IMS for improving analytical separations at low analysis times (< 30 ms).

A new trend is to integrate IMS in LC–MS workflows for achieving the separation of isomers and isobars that are not resolved in the chromatographic dimension. For example, milk oligosaccharide isomers lacto-N-hexaose (LNH) and lacto-N-neo-hexaose (LNnH) are difficult to separate by LC and tandem MS provides similar fragmentation patterns in both positive and negative mode. However, both carbohydrates are baseline resolved by IMS under negative ionization conditions [57]. In this case, the statement ‘IMS separates ions rather than molecules’ is clearly exemplified. Deprotonated forms of LNH and LNnH were separated because they presented different CCS, but it was not the case for their sodium adducts. Therefore, ionization conditions have a special relevance for IMS results and, if there are not ionization constraints (i.e., ion suppression, low ionization for targeted analytes), they must be investigated when performing IMS experiments.

In general, IMS is considered as a third separation dimension in LC–MS workflows, but it can also act as a fourth dimension if a two-dimensional (2D) chromatographic separation is performed prior to IM–MS analysis. Food matrices are highly complex and one single LC or GC separation can be insufficient to provide a satisfactory degree of resolving power for their characterization. Advanced approaches such as LC × LC and GC × GC have been developed to enhance peak capacity which is increased even more when IMS is integrated in LC × LC–MS workflows, as shown in the analysis of phenolic compounds in chestnut (i.e., ellagitannins and gallotannins), grape seed (i.e., procyanidins), rooibos tea and red wine (i.e., flavonoid and non-flavonid phenolics) [58]. TWIMS improved the practical peak capacity from 7- to 17-fold (depending on the sample) in comparison to a 2D–hydrophilic interaction liquid chromatography (HILIC)-reverse phase (RP)–LC–ToF–MS method. Regarding the orthogonality degree of separation methods, lower orthogonality was found for a HILIC–TWIMS combination (52%), whereas the orthogonality existing between RP-LC and TWIMS was surprisingly higher (73%) than the orthogonality resulted from HILIC–RP–LC combination (67%). 2D–LC–IMS–MS approaches clearly reveal a higher complexity of food samples and improve the probabilities of distinguishing between isobaric and isomeric compounds [59]. For example, 2D–ultra-high performance liquid chromatography (UHPLC)–TWIMS–ToF–MS has been applied to the characterization of ginsenosides in white and red ginsengs which are diet supplements, and from the 201 compounds identified, 10 pairs of co-eluting isobaric ginsenosides were resolved only by IMS [60]. On the other hand, 2D–GC–IMS–MS methods have not been reported for the analysis of food samples, but the potential of this strategy has already been shown for the analysis of volatiles in medical herbs [61]. Consequently, similar approaches can be applied to the determination of the VOCs fraction of foodstuff.

Finally, IMS provides structural information of the ionized compounds (i.e., CCS), giving a more complete overview of food composition [44,62]. In the context of food composition analysis, ^DT^CCSN_2_ libraries have been developed for the characterization of phenolic acids in wine [46] and lipids in bovine milk [44]. In this last case, lipid identification rates were increased when CCS, in addition to retention time and accurate *m*/*z*, was considered as molecular descriptor. Machine-learning algorithms were developed for the classification of 429 lipids according to their family class (i.e., triacylglycerides, diacylglycerides, phosphatidylcholines, and sphingomyelins) and carbon number. In general, a satisfactory classification rate (84.01%) was achieved when retention time and *m*/*z* were selected as analytical parameters. Classification accuracy was increased to 91.78% when CCS was included in the model, mainly due to the separation of isomeric species. This model was further improved to include the unsaturation level and, as a proof of concept, finally applied to the classification of 2087 bovine milk lipidomics data. As a result, 429 lipids, which were previously identified, were accurately classified whereas 179 unknown lipids were also annotated confidently. The identification of steviol glycosides represents another example of the applicability of CCS values for food characterization [62]. Steviol glycosides encompass a wide range of isomeric compounds and related molecules that can co-elute or result in similar fragment ions, impacting identification certainty. In order to improve the characterization of food products containing steviol sweeteners, a library of ^TW^CCSN_2_ values was developed and applied to the fingerprinting of 55 food commodities. ^TW^CCSN_2_ values contributed to the identification of several isomeric pairs such as rebaudioside E and rebaudioside A ([M − H]^−^
*m*/*z* 965.4230, CCS = 289.2 and 298.8 Å^2^) and stevioside and rebaudioside B ([M − H]^−^
*m*/*z* 803.3701, CCS = 269.6 and 261.2 Å^2^), without requiring ion fragmentation for identification, which cannot provide relevant information at low analyte concentrations.

### 3.2. Food Process Control

Food process control requires rapid response analytical tools, mainly intended as the analysis of many volatile compounds, in order to monitor industrial processes in near-real time and make rapid decisions if needed. In this sense, VOCs fraction can be related to storage conditions and production process, being also an indicator of the shelf life of food products. Consequently, volatiles are usually analyzed as part of food process and quality control. Stand-alone IMS represents a good option for this purpose because it allows the quick detection of volatile compounds (~20 s) at low cost, which is also attractive for industrial companies.

IMS has been widely applied to control the freshness of food products with special attention to the determination of biogenic amines [7,8], which are usually associated with fermentation processes and food degradation. Stand-alone IMS devices, which typically consist of DTIMS systems, are commercialized with different ionization sources. Therefore, the instrument to be used depends on the nature of analytes and their ionization characteristics. This is not a disadvantage for the analysis of biogenic amines by IMS because similar LODs can be reached (i.e., 0.1–1.2 ppm expressed as vapor in air for trimethylamine, putrescine and cadaverine), independent of the ionization source (i.e., photo-ionization, corona discharge, and radioactive ion sources) [63]. Stand-alone IMS methods have been reported for the determination of histamine in tuna fish [64], trimethylamine in seafood [65], and the simultaneous analysis of histamine, putrescine, cadaverine, tyramine in canned fish [66]. Stand-alone IMS applications have also been developed to detect the presence of off-flavors and contaminants generated during food processing and storage such as 2,4,6-trichloroanisole [67], and furfural and hydroxymethylfurfural toxicants [68].

Due to the low selectivity of stand-alone IMS instruments, the combination of HS–GC with IMS is recommended for the simultaneous analysis of several volatile compounds in liquid and solid samples in food quality control [5]. Despite chromatographic separations requiring several minutes, HS–GC–IMS can be less time-consuming and more environmentally friendly than other analytical and physico-chemical approaches currently used in food process control [69]. HS–GC–IMS methods have been applied to study lipid oxidation processes experienced by roasted peanuts [70] and EVOO [69] during storage.

In addition to the selective analysis of specific volatile substances, the profiling of VOCs fraction by IMS approaches can provide more complete information about the freshness and storage conditions of food products such as fish [71] and eggs [72]. VOCs profiling by GC–IMS usually requires chemometrics for data treatment in order to classify the samples, although the majority of compounds remain unidentified. It does not represent a limitation for sample classification, but compound identification is essential to understand fermentation and other decay processes. For example, up to 35 potential markers related to freshness were detected in the VOC fraction of eggs, but only a few compounds (i.e., butyl acetate, heptanal, dimethyl disulfide, dimethyl trisulfide and 1-butanol) were identified (Figure 4) [72]. Nevertheless, eggs were correctly classified according to their freshness (from 0 to 5 days at room temperature) by a principal component analysis (PCA) model. A supervised orthogonal partial least square discriminant analysis (OPLS–DA) model was finally developed to maximize the separation of both groups (i.e., fresh eggs vs. non-fresh eggs).

Volatile compounds are not only related to the shelf life of food products and their degradation. The monitoring of VOCs fraction can also be decisive in food process and quality. For example, beer fermentation is conditioned by the presence of diacetyl and 2,3-pentadione whose concentrations must be reduced below the human odor threshold. In comparison to traditional methods, the use of GC–IMS for their determination allows decreasing the analysis time from 3 h to 10 min [5]. GC-IMS has also been applied to the characterization of VOCs fraction during the fermentation of lychee beverages [73] and wine [74]. One of the advantages of GC–IMS over other analytical tools is that it enables online monitoring and control of bioprocesses [74]. However, GC separations still require several minutes, which is a disadvantage of real-time decisions. By contrast, IM–MS analyses take a few milliseconds allowing the direct monitoring of VOCs evolution, which involves time- and cost-saving on sample preparation as well as avoids process interruption. Corona discharge–DTIMS–ToF–MS has been proposed for the on-line monitoring of VOCs formation in coffee roasting processes [75]. More than 150 VOCs were observed during the roasting of Brazilian *Coffea arabica* beans, and several alkyl pyrazines, fatty acids and other organic acids were identified. Some of them were isomers and isobars, so MS was not enough to identify them. Nevertheless, the integration of DTIMS in the analytical workflow allowed their separation according to their ^DT^CCSN_2_ (e.g., unsaturated fatty acids presented smaller ^DT^CCSN_2_), facilitating their identification.

### 3.3. Food Authentication

Chemical fingerprinting of food products and chemometrics are widely used for food authentication and ultimately to identify food fraud [76]. Within this framework, metabolomics fingerprinting is a promising strategy for food authentication that combines both approaches. In this sense, it is very efficient and has already shown clear benefits over traditional methods [77]. At the moment, only one study has been reported about the applicability of LC–IM–MS in metabolomics fingerprinting for food authentication [78]. The metabolic fingerprint of 42 red wines from the Republic of Macedonia was obtained by LC–ESI–DTIMS–ToF–MS, and the detected features were characterized in terms of retention time, accurate mass and ^DT^CCSN_2_ under positive and negative ionization conditions. ^DT^CCSN_2_ values added an extra-identification point for the putative identification of several phenolic compounds and other grape reaction products. After data treatment by PCA, Vranec wines were clearly distinguished from other varieties such as Cabernet Sauvignon or Merlot wines.

In general, IMS-based methods for food authentication have mainly been developed for the fingerprinting of volatile and semi-volatile compounds, usually involving GC–IMS hyphenation. GC–IMS methods in combination with chemometrics have been successfully applied to the authentication assessment of oils [79,80], meat products [81,82], wines [83] and honey [84]. For example, EVOO are ‘designation of origin’ (DO) products of high value, so their authentication is required not only to detect any fraud related to oil quality but also to identify mislabeling regarding origin. Based on the analysis of the VOCs by HS–GC–DTIMS, high discrimination rates (i.e., 98% and 92% by PCA-linear discriminant analysis (LDA) and PCA-k-nearest neighbors (kNN), respectively) have been reached to effectively discriminate between EVOO from Italy and Spain [80]. A similar HS–GC–DTIMS approach has been followed to authenticate Iberian hams and discard mislabeling [81]. The monitoring of specific non-identified markers by HD–GC–DTIMS and the application of orthogonal projections to latent structures discriminant analysis allowed to distinguish hams with a discrimination rate of 100%. As a third example, a HS–GC–DTIMS method has also been applied to identify the botanical source of different types of honey [84], since faulty declaration of the botanical source constitutes one of the main frauds on honey products. VOC profiling by HS–GC–DTIMS provided a discrimination rate (> 98.6% according to PCA-linear-discriminant analysis (LDA)) as high as proton nuclear magnetic resonance (^1^H-NMR) spectroscopy, which has been widely investigated for honey authentication in the last years.

Despite the efficiency showed by GC–IMS for the authentication of food products, the information resulting from GC–IMS analyses is very limited since MS is not included in the analytical workflow and markers cannot be identified. Only those compounds whose standards are available in the laboratory can be identified according to their retention and drift time, whereas the majority of compounds in the sample remain unknown. Nevertheless, discrimination of food products of different geographical origin, nature, quality, etc. can be achieved by GC–IMS–chemometrics methods based on unidentified markers. These strategies are usually recommended as screening approaches since molecular identities are generally required for confirmation.

### 3.4. Food Adulteration

The detection of food adulteration represents a particular case of food authentication in which substances are intentionally added to food products to decrease and mask their quality or valuable ingredients are removed in order to obtain higher economic profit. Consequently, similar IMS-based approaches as those shown in Section 3.3 have been proposed for the identification of this type of food fraud. HS–GC–IMS in combination with chemometrics has been evaluated for detecting the adulteration of winter honey (derived from *Schefflera actinophylla* (Endl.) *Harms* and wild *Eurya* spp. (Theaceae)) with cheaper *Sapium* honey [85], canola oil with other vegetable oils (i.e., sunflower, soybean, and peanut oils) [86], and crude palm oil with process byproducts of lower quality (i.e., palm fiber oil and sludge palm oil) [87]. As previously mentioned, GC–IMS methods should mainly be used as screening approaches because MS results essential for confirmation. However, screening methods are required for rapid analysis at lower cost and are of special interest for carrying out in situ analyses. Thus, only a few suspicious samples are sent to the laboratory for confirmation. In this context, stand-alone in combination with chemometrics has been investigated for the rapid detection of EVOO [88], sesame oil [89], and flaxseed oil [90] adulteration with vegetable oils of lower quality. Therefore, and despite the lack of selectivity of IMS, these applications demonstrate its efficiency to initially discriminate between allegedly adulterated samples and samples that are in compliance with their quality and labeling.

### 3.5. Chemical Food Safety

Chemical food safety covers the determination of a wide range of residues and contaminants (i.e., pesticides, veterinary drugs, toxins, environmental contaminants, etc.) in feed and food-related matrices; being one of the food science fields where IMS has found a wider application. Within this framework, the analysis of pesticides in a great variety of vegetables (i.e., apples, tomatoes, cucumbers, etc.), juices, oils, animal feed and water samples has been the topic most investigated [8]. IMS has been shown to be a solution for answering current society’s concerns such as the presence of glyphosate in drinking water (as well as in other food products). The determination of glyphosate is quite difficult due to its ionic character, low volatility and low molecular weight. However, the LOD achieved by IMS for drinking water analysis (i.e., 10 μg/L) is comparable to those reported by HPLC and GC–MS (i.e., 0.02–50 μg/L) or ion chromatography (i.e., 15.4 μg/L) methods [91]. Despite of the potential of IMS for food safety applications, other food contaminants and residues rather than pesticides have been scarcely studied by IMS [8].

The analysis of residues and contaminants in food products faces the same issues as other areas of food analysis (i.e., high complexity of food matrices requiring high selective analytical techniques, presence of isomers and isobars, and requirement of detecting compounds at very low concentration levels) but, in this particular case, it must also comply several regulations according to the compounds analyzed. From a legal perspective, the employment of MS is practically mandatory in order to reach enough identification points (IP), so IM–MS related methods are of major interest for chemical food safety applications. However, IMS-based methods do not follow current guidelines and regulations concerning analytical methods intended to the determination of pesticides (e.g., SANTE/11813/2017 [92]), veterinary drugs (e.g., Regulation 2002/657/EC [93] or mycotoxins (e.g., SANTE/12089/2016 [94]). Indeed, because this particular technique has not been considered in the corresponding documents and no IP is allocated to CCS determination yet, this impairs its use as official identification criteria. For instance, scientists currently working is the revision of decision 2002/657/EC are debating the value of this new parameter as a criterion for identification and, if necessary, the value of the IP awarded [(acceded on 12 June 2019), [95]].

Under this context, stand-alone IMS and GC–IMS methods have been widely proposed for chemical food safety analyses [8], but they are outside the current frame of food law enforcement. Moreover, despite the use of stand-alone IMS has been shown to be an effective strategy for the rapid and in situ detection of residues and contaminants [96], its selectivity is limited and samples must normally be submitted to selective sample treatments prior analysis. In this sense, molecularly imprinted solid-phase extraction (MIPSE) and immunoaffinity chromatography (IAC) have been commonly employed for the selective extraction of the analytes of interest and to avoid matrix background. For example, IAC has been applied to the analysis of fungicides in strawberry juices and wines [97] and mycotoxins (i.e., aflatoxins B1 and B2) in pistachios [98], whereas MISPE has been use to study the uptake and translocation of a neonicotinoid pesticide (i.e., imidacloprid) in chili and tomato plants [99]. Regarding GC–IMS methods, IMS has been useful to separate pesticides that were not resolved in the chromatographic dimension [100,101], although in general it merely acts as detector [32].

In chemical food safety, tandem mass spectrometry (MS/MS) and high-resolution mass spectrometry (HRMS) are usually employed with the aim of achieving the unequivocal identification of compounds. The combination of IMS and HRMS or MS/MS has commonly been applied to clean up mass spectra by removing matrix interferences and signal background. As a result, the task of mass spectra interpretation is reduced and compounds at lower concentrations are detected more easily. FAIMS and DMS are ion filters and this characteristic has been exploited to reduce the signal background in the determination of the mycotoxin zearalenone and its metabolites in cornmeal [102]. LODs were consequently improved up to 25 and 42.5-fold when applying FAIMS–ESI–MS instead of ESI–MS/MS or ESI–MS, respectively. Despite the characteristic of ion filtering is usually assigned to FAIMS/DMS technology, DTIMS and TWIMS are also able to isolate analyte signals from chemical background, improving concentration sensitivity [30,103]. In the context of chemical food safety, signal to noise ratio (S/N) of LC–ESI–ToF–MS methods intended to monitoring steroids and their metabolites in livestock were increased between 2 and 7 fold by the integration of TWIMS in the analytical workflow (Figure 5) [30].

On the other hand, IMS–ESI–MS methods provide rapid throughput as observed in the separation of isomeric perfluoroalkyl substances (PFAS) by DMS–trap quadrupole (LTQ)–MS [104]. Nevertheless, food samples are very complex and matrix compounds can cause ion suppression when applying ESI ionization. This issue has a great impact on signal sensitivity so it must be avoided for achieving robust and confident analyses. Thus, samples must normally be submitted to LC (or GC, SFC, CE) separation after specific sample treatments and prior to IM–MS analysis in order to overcome this effect. For example, FAIMS coupled to LTQ–Orbitrap has been shown to be a useful tool to separate paralytic shellfish toxins epimeric pairs [105], but their analysis in shellfish tissue extracts required a previous HILIC separation due to severe ionization suppression from matrix components. In general, 3D–LC–IM–MS separation is usually recommended when IMS is applied within the framework of food safety.

In this sense, LC and IMS are complementary separation techniques because they are based on different separation principles. Saxitoxins, which are a class of marine neurotoxins present in shellfish, comprise a group of isomers that are not differentiated by MS. Some diastereomers can be separated by HILIC but it does not distinguish between non-sulphated saxitoxins analogues. These molecules can be separated by TWIMS which makes HILIC–TWIMS–MS the best solution for the analysis of these group of substances in one single run [106]. Other examples of isobars and isomers separation by IMS include veterinary drugs such as ractopamine/isoxuprine ([M + H]^+^; *m*/*z* 302.1751, ^TW^CCSN_2_ = 171.9 Å^2^ and *m*/*z* 302.2025, ^TW^CCSN_2_ = 173.6 Å^2^, respectively) by TWIMS [107] and isomeric environmental contaminants such as 2,2′,4,5′,6-pentachlorobiphenyl (PCB-103) and 3,3′,4,4′,5-pentachlorobiphenyl (PCB-126) ([M − Cl + O]^−^; *m*/*z* 304.9095, ^DT^CCSN_2_ = 160.7 and 164.4 Å^2^, respectively) by DTIMS [108]. Furthermore, if the separation of isobars and isomers cannot be achieved under standard conditions in the drift cell (i.e., only containing the drift gas), organic solvents (e.g., acetonitrile, methanol, isopropanol, etc.) can be added into the buffer gas in DMS and FAIMS systems in order to improve ion mobility separation [109]. This approach has been followed for the separation of the neurotoxin β-*N*-methylamino-l-alanine (BMAA) and its isomer β-amino-N-methylalanine (BAMA) in mussel tissues [110]. Both isomers were only resolved by HILIC–DMS–QqQ/LIT–MS when 0.35% acetonitrile was added in the DMS carrier gas. DMS also improved the low sensitivity traditionally observed for the analysis of BMAA by LC–MS methods.

Finally, as previously mentioned, IMS also provides additional analytical information to support molecular assignment. For example, drift time has been used to identify 100 pesticides in different vegetables and fruits by LC–TWIMS–ToF–MS [111]. This analytical property is not influenced by the matrix but drift times are instrument-dependent, so these data cannot be extrapolated to other IMS platforms. For this reason, the use of the CCS as the information provided by IMS has been extended. Consequently, CCS databases can be created and applied to support the determination of compounds, reducing the number of false negative/positives found in classical LC–MS workflows [112], and increasing the detection rates at low concentration levels of residues [39]. In the current context of Regulation 2002/657/EC revision, it has emerged the proposal to include the CCS as IP for the identification of veterinary drugs in food [95]. Scientists and experts are currently debating the value of grating IPs to the CCS and, if so, to determine its contribution in the 4 and 5 IPs required for the confirmation of substances with maximum residues limits (MRLs) and non-authorized substances, respectively. However, several concerns have been raised such as the lack of CCS databases and the tolerance accepted for CCS measurements. While, until a few years ago, CCS had been barely used as verification parameter in screening approaches due to the lack of databases, this situation has now recently changed and several CCS databases for pesticides [112,113], veterinary drugs [114,115,116], mycotoxins [117] and other contaminants [108], have been reported in the last five years.

Finally, recent applications of LC–ESI–TWIMS–ToF–MS methods, which include the CCS as signal filter in data processing, report the analysis of mycotoxins in cereals [117] and pesticides in green tea powder, fresh garlic, leek, fresh herb chives and rye [112], fish feed [31], and in 20 food products (e.g., strawberries, honey, chia seed, etc.) that belong to six different commodities according to SANTE/11813/2017 [39]. Other LC-TWIMS-MS applications in the context of food safety encompass the identification and structural characterization of residues and contaminants, either for evaluating exposure [118] or for studying their biotransformation [119].

## 4. Current Perspectives of Ion Mobility Spectrometry

Significant progress has been made in IMS technology in the last years, especially in IM–MS hyphenation. This fact has been directly reflected in the number of IMS-related applications that are currently published in various scientific fields including food science, and the number of related papers will keep growing due to technological developments in this analytical technique. Advances are expected in three different senses: IMS coupling with MS technologies offering higher resolution than ToF-MS (i.e., Fourier transform ion cyclotron resonance (FTICR) and Orbitrap–MS), improvement in IMS resolution, and implementation of the CCS as a parameter to support molecular assignment. In this section, only the last two topics will be discussed in more detail because the first is more related to improvements in MS acquisition rates than to IMS evolution. Several prototypes based on DTIMS/TWIMS–FTICR/Orbitrap–MS hyphenation have already been proposed; so further developments are expected [41]. These developments will certainly open up new possibilities for enhanced food fingerprinting, compounds identification and, ultimately, the discovery of substances, including contaminants, present in food but not detected until now.

### 4.1. Improvement in Peak Resolution

The resolving power (R_p_) of IMS is typically expressed in the scale of CCS for IMS-platform comparison, although it excludes FAIMS/DMS systems. R_p_ is calculated according to Equation (6) where ΔCCS represents the full width of the peak at half its maximum height (FWHM).
(6)$$R_{p}=\frac{\mathrm{CCS}}{\Delta \mathrm{CCS}}$$

As a consequence of the R_p_ of IMS, the integration of this analytical technique in LC–MS workflows increase their peak capacity. Thus, it can be expected that LC-IM-MS will quickly replace LC–MS methods in many applications, including food analysis. Current IMS instrumentation provide a R_p_ up to 300, so molecules presenting CCS differences (Equation (7)) as small as 0.5% can be separated [120]. However, commercial systems are not able to routinely offer this type of performance. TIMS is the only capable to provide a R_p_ higher than 200 [41], which is considered the lower limit of ultra-high resolution (UHR)IMS [23]. R_p_ of DTIMS systems is around 100 whereas TWIMS instruments currently reach a maximum R_p_ of 40 [14,23]. Therefore, the development of UHRIMS and its implementation in routine analysis obviously represent some of the main challenges of IMS, but it will also contribute to extend the use of this technique.
(7)$$\Delta {\mathrm{CCS}}_{A,B}\left(\%\right)=\frac{{\mathrm{CCS}}_{B}-{\mathrm{CCS}}_{A}}{{\mathrm{average}\;\mathrm{CCS}}_{A,B}}\times 100$$

As a result of the improvements on R_p_ expected in UHRIMS technology, IMS separations will be able to provide similar peak capacities as LC at lower analysis times, which will be revolutionary from an analytical point of view and transformative for several applications [121]. In general, limitations on the applied field involve that R_p_ can only be increased by extending the drift path, although it presents some practical constraints [17,23]. In this sense, two IMS approaches (i.e., cyclic-TWIMS and structure for lossless ion manipulations (SLIM)–TWIMS) have successfully overcome these drawbacks and, consequently, are gaining attention due to their enhanced R_p_. In addition, a third alternative was previously explored, namely the high resolution ion cyclotron mobility spectrometry (cyclic-DTIMS) [122], which is a circular 180.88 cm-length drift tube consisting of four quarter-circle drift tubes with ion funnels in between to re-focus the ions. Ions undergo multiple passes through the drift path in order to improve their separation. Successive improved versions of this technology have been shown to reach a R_p_ in excess of 1000 (as a consequence of 100 transits or cycles and involving a drift length of over 180 m) [123]. Nevertheless, no further research has been reported about cyclic-DTIMS, probably due to the long measurement times required and the significant loss of ions at long drift times of more than a second [23].

In cyclic-TWIMS, the drift cell is arranged in a circle configuration with pre- and post-store cells for ejecting the ions in and out [124,125]. As a result, ions can be submitted to several passes enhancing their separation. The resolution of multiple passes is given by Equation (8) where A is the single pass resolution (~65, 98 cm pass length), n represents the number of passes, and z is the ion charge state.
(8)$$R_{P}=A{\left(\mathrm{nz}\right)}^{1/2}$$

One single separation can be enough to obtain a general overview of samples and select the ATD regions of interest that will be submitted to multiple passes. The selection of ATD regions is required in order to avoid that the fastest ions trap the slowest ones, causing a ‘wrapping effect’. Due to its enhanced R_p_, cyclic-TWIMS provides new insights of ions conformation not previously described, as observed for the three protomeric species of the veterinary drug danofloxacin (Figure 6) [126]. As discussed above, it is relevant to characterize protomers in the food safety context because they do not fragment in the same way and results could be non-compliant with confirmation criteria based on ion fragmentation ratios. This technology was recently launched to the market by Waters Corporation during the last American Society for Mass Spectrometry (ASMS) meeting (2019).

SLIM–TWIMS technology is not yet commercially available, but it is also catching the attention due to its high R_p_ resulted from its long drift path (~337, 540 m; 40 passes, 13.5-m path) [121]. As occurs with cyclic-IMS, SLIM–IMS can be based on DTIMS or TWIMS separations. However, TWIMS technology is usually applied because it allows longer path lengths than DTIMS, which presents voltage limitations [23]. SLIM–TWIMS systems consist of two planar surfaces fabricated using printed circuit boards and containing an array of electrodes. Ions travel through a serpentine path and are confined due to the application of DC and RF potentials [127]. SLIM–TWIMS offers a longer drift path than cyclic-TWIMS and, as a consequence, a larger range of mobilities can be monitored simultaneously. Although SLIM–TWIMS approaches have not been directly applied to food analysis, it has been shown to improved separations of lipid and peptide isomers [128] as well as glycans that differ in CCS by as little as 0.2% [129], which can have potential applicability in food composition analysis and food fingerprinting.

### 4.2. Implementation of CCS in Current Analytical Workflows

The CCS parameter is closely correlated to *m*/*z*, but its use as molecular descriptor in traditional analytical workflows has been shown to be effective in improving data processing by reducing false positive/negative results [112,113]. Despite its potential to support compound identification, the lack of CCS libraries is considered to be the major drawback for reaching this goal and implementing the CCS parameter in current analytical workflows [8]. In an effort to overcome this issue, several CCS databases have been reported in the last years [15], although they are not enough considering the number of molecules that remain uncharacterized. In order to extend the knowledge provided by CCS databases created experimentally, other strategies such as computational modeling and, more recently, machine-learning based prediction have also been investigated [130]. Machine-learning approaches are very effective for CCS prediction and are becoming very popular for generating CCS values of molecules on a large scale because their application only takes a few seconds/minutes whereas computational modeling is computationally intensive. For example, a machine-learning model based on artificial neural networks (ANNs) has been developed for the CCS prediction of small molecules selecting 205 compounds for model training, verification and blind test sets (ratio 68:16:16) [131]. This model was subsequently applied to the analysis of ten pesticides in spinach samples, and deviations between the observed and predicted CCS values for the protonated ions of pesticides were smaller than 5.3%.

CCS databases are typically based on one-platform measurements and, taking into account that CCSs are conditional values, the implementation of current libraries in other IMS platforms already introduces uncertainty into CCS measurements [20]. Theoretically, CCS values are platform-independent but they are influenced by several experimental parameters (i.e., E/N, temperature, and buffer gas) as indicated above. Specifically, the nature of buffer gas has a great influence on the CCS of molecules [132], and CCS values measured in different drift gases cannot be directly compared, as shown for a wide variety of pesticides whose CCSs have been obtained in He, CO_2_, N_2_O and SF_6_ by DTIMS [133]. Within this framework, it may actually be more appropriate to develop standardized CCS libraries based on multiple platform measurements. Inter-platform studies should also consider different IMS forms (i.e., DTIMS, TWIMS, TIMS) to establish if standardized CCSs can be implemented in all of them or whether, on the contrary, ^DT^CCS, ^TW^CCS and ^TIMS^CCS values must be reported and used. In general, similar CCS values are provided by these three IMS technologies when using the same buffer gas as observed for the sodium adduct of 25-hydroxyvitamin D3 [134]. However, another recent study about the CCS of 35 pharmaceuticals, 64 pesticides, and 25 metabolites of pesticides has shown that, although the majority of ions present differences lower than 1.1% between ^DT^CCS and ^TW^CCS values, both values cannot be compared in all cases [135]. Deviations up to 6.2% between ^DT^CCS and ^TW^CCS values were observed for several ions.

In the same vein, there is still a knowledge gap in the use of CCS databases and the variability associated with CCS measurements. Recent studies have shown high repeatability over time [30,112], and negligible impact of sample matrix on CCS values [39,113]. Despite this, the use of this parameter for specific analytical applications regulated by guidelines requires a comprehensive assessment of reproducibility across different laboratories and instrument types. Inter-laboratory reproducibility has barely been evaluated, and only few studies have tackled this issue [22,112,136]. This information is highly relevant to establish confidence intervals for CCS measurements and thresholds for their comparison with CCS values in databases. An inter-laboratory study involving four DTIMS-platforms have recently reported absolute bias of 0.54% to the standardized stepped field ^DT^CCSN_2_ values on the reference system [22]. Under this context, the current threshold of 2% accepted for CCS measurements seems to be too wide and could be reduced. As a result, the number of possibilities for peak assignment will be decreased when using the CCS for compound identification. Based on current evidence, this threshold could potentially be reduced to at least 1–1.5% [30,112,113]. Looking ahead to the implementation of the CCS parameter to support compound identification, a score system based on deviations from standardized CCS values should also be developed. Scoring systems based on mass spectra are typically used for the putative identification of molecules, so a similar approach could be suitable for the integration of CCS values in data processing workflows.

In addition to the standardization of CCS libraries and establishment of thresholds for CCS measurements, there is a third issue that requires further development. CCS calibration must be carried out in DTIMS (in single-field mode), TWIMS and TIMS systems before CCS measurements, and must be performed under the same operational conditions applied to obtain the CCS of analytes. Until now, there is neither consensus on the application of standardized calibration protocols nor on the application for this purpose of primary standards, which also serve for preparing reference materials [20]. A protocol has been reported for TWIMS calibration and operation [137], but different calibration procedures and calibrants are applied in other IMS systems. Under this context, ‘Agilent tunemix calibration standard’ (i.e., mixture of hexakis-fluoropropoxyphosphazines) [22], poly-DL-alanine [107], and ‘major mix IMS/ToF calibration kit’ from Waters (i.e., mixture of small molecules such as caffeine or sulfadimethoxine, and including poly-DL-alanine) [116] are currently widely used as CCS calibrants. Other alternative CCS calibrants such as the dextran have also been investigated for obtaining the CCS of carbohydrates [138]. Therefore, more studies about CCS calibration are still needed since, in the case of some classes of compounds, the accuracy of CCS measurements depends on the chemical nature of the calibrant [139]. Within this framework, both standardized calibration procedures and primary standards of different chemical class seem to be crucial in order to increase confidence in CCS comparison and measurements. The development of reference materials and standardized calibration protocols will contribute to the implementation of the CCS, but it calls for a great effort from IMS community including IMS suppliers.

## 5. Conclusions and Perspectives in Food Analysis

The implementation of IMS in food science is quite new but its applicability is growing exponentially. Nowadays, there is a wide range of methods intended for food composition, process control, authentication, adulteration and safety analysis, covering a great variety of molecules (i.e., macronutrients such as proteins, lipids, carbohydrates, vitamins; micronutrients such as polyphenols; process and biodegradation products such as biogenic amines; residues such as pesticides and veterinary drugs; or contaminants such as toxins and environmental pollutants), and involving different IMS forms (i.e., DTIMS, TWIMS, TIMS, FAIMS and DMS).

Stand-alone IMS instruments are portable and provide a quick response (< 30 ms), so they have been shown to be very efficient for in situ analysis and real-monitoring, such as in food process control where rapid decisions have to be made. It can be expected that miniaturized and portable IMS instruments will be implemented for food analysis as already occurs in the analysis of chemical warfare agents and drugs in airports, courts, etc. Potential applications are in situ monitoring of ripening processes of crops, depletion of residues after veterinary treatments in farms, etc.

In many applications, stand-alone IMS approaches offer low selectivity, so they are limited to the determination of specific compounds and usually require exhaustive sample treatments prior analysis due to the complexity of food matrices. In order to improve selectivity, IMS is commonly coupled to a chromatographic technique, especially GC if MS detection is not used. GC–IMS methods are very popular for the analysis of VOC fraction of food. This approach is widely used in food authentication where HS analysis is typically carried out. HS–GC–IMS in combination with chemometrics is broadly applied for food fingerprinting and product discrimination according to their quality, origin, etc. However, this strategy barely exploits the potential of the IMS technique (since it usually acts as simple detector) and is limited by the low number of compounds that are identified (i.e., only substances with standards available in the laboratory are identified based on their retention and drift times). The development of GC–IM–MS strategies will give more knowledge about VOCs composition, overcoming the current boundaries of GC–IMS approaches for food characterization and authentication. Consequently, for example, food fermentation and decay processes will be better understood since mass spectra will be obtained.

The recent commercialization of hyphenated IM–MS instruments, usually as part of LC–IM–MS platforms, has been the reason why IMS is becoming very popular in food science. IMS is extending the current boundaries of LC–IM–MS methods by introducing an extra separation dimension that allows the separation of isobars and isomers (not always separated in the chromatographic dimension and undistinguished by MS) and the isolation of analytes of interest from chemical background (so improving S/N and enhancing sensitivity). Although the correlation existing between IMS and MS has been a topic of discussion for a long time, the applications included in this review show that a higher number of compounds are detected by LC–IMS–MS in comparison to traditional LC–MS workflows, which gives a more complete picture for food characterization. Moreover, the integration of IMS provides cleaned-up chromatograms and mass spectra facilitating data interpretation.

In addition, IMS gives additional information to retention index and mass spectra of molecules (i.e., K_0_, drift time, CV and/or CCS) which can ultimately be used for compound identification. Within this framework, the CCS of molecules has acquired great relevance. Despite this parameter is correlated to *m*/*z*, it has been shown to be useful to distinguish compounds with different chemical nature, but also to differentiate among close chemical molecules. From a regulatory point of view, the application of the CCS parameter for the identification of residues and contaminants in food is currently under discussion. Nowadays, the development of open-access CCS databases (based on inter-laboratory measurements for proposing normalized reference values), the requirement of standardized calibration procedures and calibrants for CCS measurements, and the establishment of thresholds for CCS measurements, are viewed as the main challenges to tackle for the implementation of CCS in food analysis. Based on current knowledge and trends, CCS will definitely be included in current analytical workflows to support food characterization and, probably, legally accepted as complementary information to confirm the presence or absence of residues and contaminants in food products.

Technological developments experienced by IMS, and especially by IM–MS hyphenation, are going to have a great impact on the implementation of this technique in the food science field. Enhancements in R_p_, currently lead by cyclic-TWIMS and SLIM–TWIMS technologies, will contribute to overcome the challenges arise from the complexity of food matrices (i.e., high number of compounds with different chemical nature at different concentration levels). Under this context, they will enable the discovery of unknown food components with bioactive properties which are normally at low concentration levels and are typically masked by isobaric and isomeric compounds or by food major components. In addition, more complete fingerprints of food products will be achieved, providing more detailed information for food authentication and detection of food adulteration.

Finally, the integration of IMS in current LC–MS (or GC–MS, CE–MS, SFC–MS) workflows involves certain technical challenges for operators already working in food analysis and for whom this technique is generally unknown. The addition of a third dimension also involves more complex data. This issue hinders data interpretation, especially in the case of food fingerprinting where large datasets are generated. Therefore, the development of simple benchtop LC–IM–MS platforms and user-friendly software for IMS operation and data treatment will be crucial to the success of the implementation of IMS in food analysis.

## Figures and Tables

**Figure 1 molecules-24-02706-f001:**
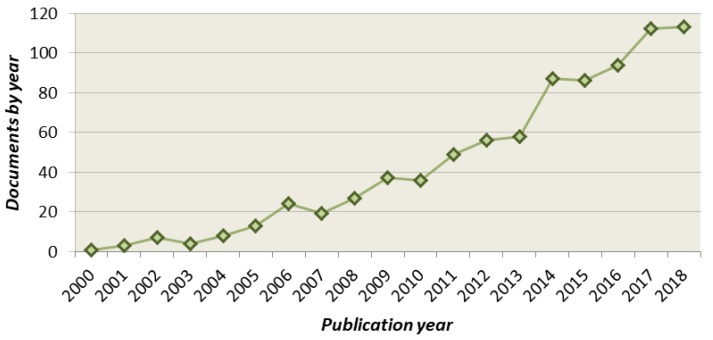
Search of literature related to ion mobility spectrometry (IMS) in food analysis from 2000 to 2018 on Scopus database. The terms “ion mobility spectrometry” and “food” have been included in the search topic.

**Figure 2 molecules-24-02706-f002:**
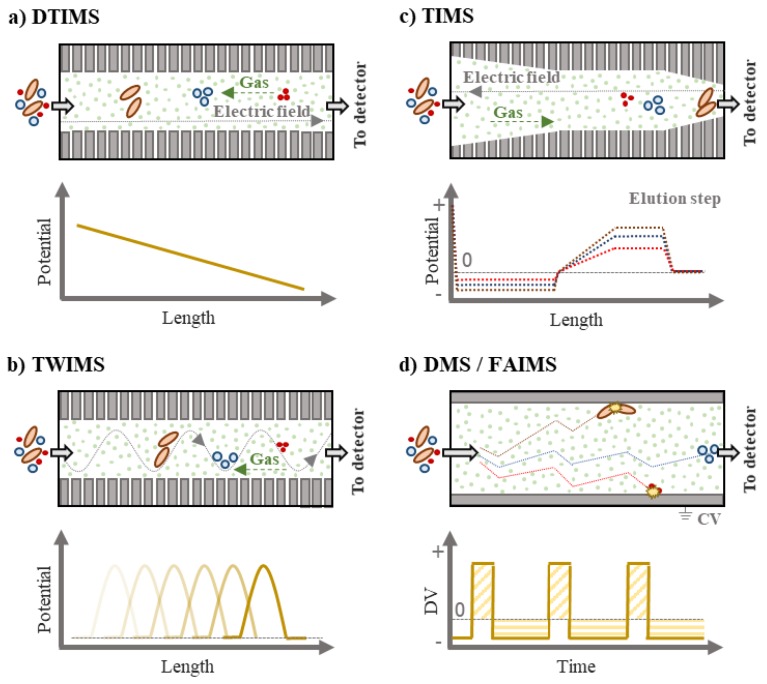
Schematic representation of commercially available IMS forms.

**Figure 4 molecules-24-02706-f004:**
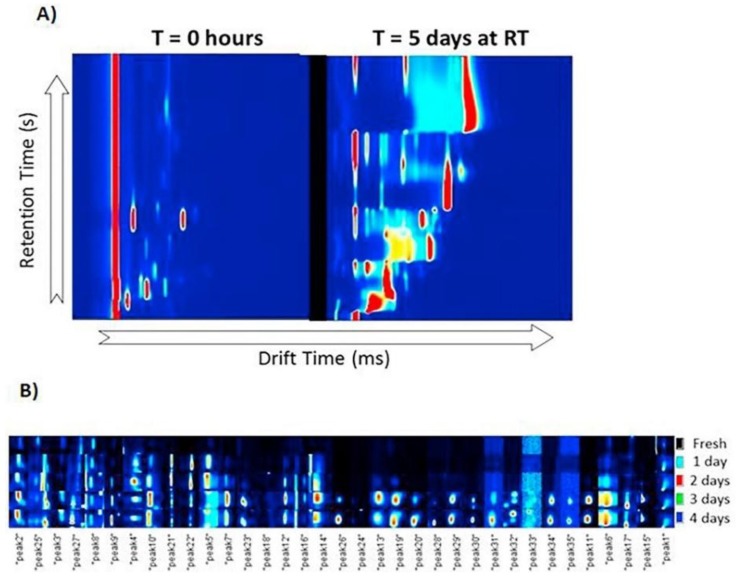
(**A**) Ion mobility spectrum of an egg product at T = 0 h (**left**) and after 5 days at room temperature (**right**). The red line identifies the reaction ion peak (RIP) position. (**B**) Global overview of the spots or ‘features’ identified in one egg product at different time points (from 0 to 4 days). Figure reprinted with permission from [72]. Copyright (2019) Elsevier.

**Figure 5 molecules-24-02706-f005:**
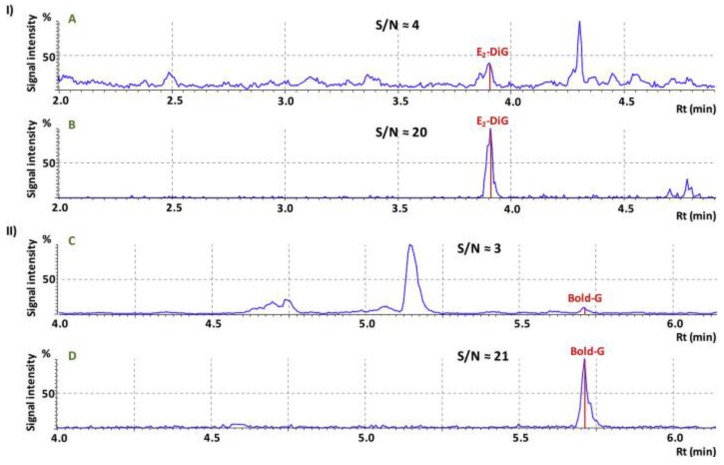
Extracted ion chromatograms (EICs) resulted from the analysis of: I) estradiol diglucuronide (E_2_-DiG; 2 μg mL^−1^; [M + Na]^+^), and II) boldenone glucuronide (Bold-G; 0.2 μg mL^−1^; [M − H]^−^) in adult bovine urine samples. The following filters were applied for signal processing of related total ion chromatograms: A) *m*/*z* 647, B) *m*/*z* 647 and drift time range between 11.3 and 11.7 ms, C) *m*/*z* 461, D) *m*/*z* 461 and drift time range between 4.9 and 5.2 ms. Figure adapted from [30], which is licensed under CC BY-NC-ND 4.0 (changes: example III has been removed).

**Figure 6 molecules-24-02706-f006:**
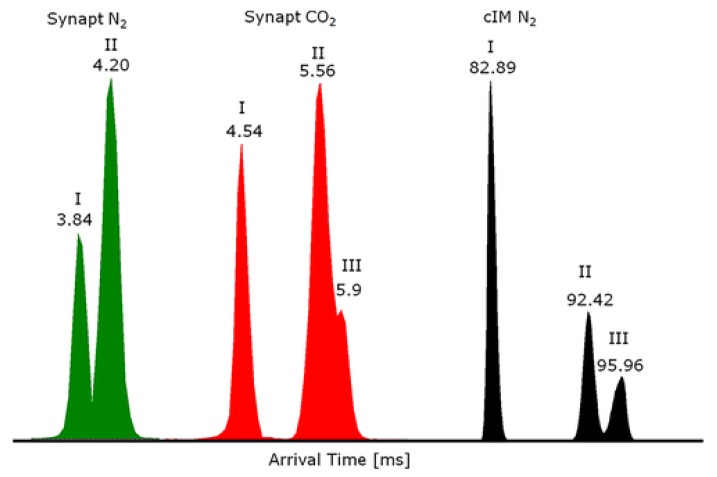
Danofloxacin IMS protomer separation using the Synapt (N_2_ and CO_2_ IMS gas) and cyclic-IMS (N_2_ IMS gas, 5 passes) systems. Figure reprinted with permission from [126]. Copyright (2019) Wiley.

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
