# Peer review of "Ion Mobility Spectrometry in Food Analysis: Principles, Current Applications and Future Trends"

_molecules, 2019, doi:10.3390/molecules24152706_

Round 1

Reviewer 1 Report

This review presents a general overview of IMS principles and the current state of the art of this technology for food analysis purposes. The subject of this review article is very interesting. The article is well-organized and includes the main literatures published in recent years. It is of great significance to understand the research status of IMS in food analysis. One suggestion: the author should discuss in detail the challenges and future trend of IMS in food analysis.

Author Response

We thank the reviewer for his/her nice comments. Challenges and future trends related to food analysis by IMS have been discussed in more detail in Section 'Conclusions', as suggested by the reviewer. In this sense, this section has been renamed as 'Conclusions and perspectives in food analysis'.

Reviewer 2 Report

This review is an interesting work which will be very useful for users of IMS. I recommend summarizing the manuscript since there are some comments that are repeating along the manuscript and to avoid comments or references which are not related to IMS papers such as [2-4]. The information included in section 2 can be summarized since it can be found in other recent reviews. In section 2.1, G.A.S. company should be included. Please add the name of the authors in reference 57. Section 4 does not fit well under the title of the manuscript. I should recommend to summarize it or even delete this section.

Author Response

First of all, we would like to thank the reviewer for his/her suggestions.

- As suggested by the reviewer, some sentences and paragraphs have been removed from the manuscript in order to summarize it, avoid the repetition of ideas as well as comments and/or references (e.g. [64-65], [72-74], etc.) that are not directly related to IMS. 

- Regarding sections 2 and 4, which are related to IMS principles and perspectives, we prefer to keep them as they are in the original version of the manuscript. It is true that there are several recent reviews about this topic, but they can be too technical for people who is not familiar with IMS. IMS is becoming popular within food chemistry field, but many researchers in the area are not familiar with this technique yet. For this reason, we have decided to write this review, not only to show them the current applications of this techinque in food analysis, but also provide them the basic principles of IMS in a simple way. This issue is already stated in Section 1: 'this review provides a general overview of IMS principles and presents the current state of the art of this technology for food analysis purposes'.

Consequently, the title of the manuscript has been modified in order to cover the overview of IMS principles and perspectives.

- G.A.S. company has been included in Section 2.1.

- The name of authors in reference 57 has been included.